# Pharmacogenetic Variants Associated with Fluoxetine Pharmacokinetics from a Bioequivalence Study in Healthy Subjects

**DOI:** 10.3390/jpm13091352

**Published:** 2023-09-01

**Authors:** Carlos Alejandro Díaz-Tufinio, José Antonio Palma-Aguirre, Vanessa Gonzalez-Covarrubias

**Affiliations:** 1Tecnologico de Monterrey, School of Engineering and Sciences, Mexico City 14380, Mexico; cdiaztuf@gmail.com; 2Axis Clinicals Latina, Mexico City 07870, Mexico; palma.a@axisclinicals.com; 3Laboratorio de Farmacogenomica, Instituto Nacional de Medicina Genomica (INMEGEN), Mexico City 14610, Mexico

**Keywords:** fluoxetine, pharmacokinetics, metabolism, genotyping, bioequivalence

## Abstract

Fluoxetine is one of the most prescribed antidepressants, yet it still faces challenges due to high intersubject variability in patient response. Mainly metabolized by the highly polymorphic gene *CYP2D6*, important differences in plasma concentrations after the same doses are found among individuals. This study investigated the association of fluoxetine pharmacokinetics (PK) with pharmacogenetic variants. A bioequivalence crossover trial (two sequences, two periods) was conducted with fluoxetine 20 mg capsules, in 24 healthy subjects. Blood samples for fluoxetine determination were collected up to 72 h post-dose. Subjects were genotyped and single nucleotide variants (SNV) were selected using a candidate gene approach, and then associated with the PK parameters. Bioequivalence was confirmed for the test formulation. We found 34 SNV on 10 genes with a quantifiable impact on the PK of fluoxetine in the randomized controlled trial. Out of those, 29 SNVs belong to 7 CYPs (*CYP1A2*, *CYP2B6*, *CYP2C9*, *CYP2C19*, *CYP2D6*, *CYP3A4*, *CYP3A5*), and 5 SNVs to 3 genes impacting the pharmacodynamics and efficacy of fluoxetine (*SLC6A4*, *TPH1*, *ABCB1*). Moreover, decreased/no function SNVs of *CYP2D6* (rs1065852, rs28371703, rs1135840) and *CYP2C19* (rs12769205) were confirmed phenotypically. Our research contributes to deepening the catalog of genotype-phenotype associations in pharmacokinetics, aiming to increase pharmacogenomics knowledge for rational treatment schemes of antidepressants.

## 1. Introduction

Depression is a common mental disorder, with 5.0% of adults affected worldwide and 5.7% among the elderly [1]. “Depression” references in scientific documents have increased exponentially since 1950, and documents mentioning “selective serotonin reuptake inhibitors (SSRIs)” have increased at a rate of approximately 41 per year in the past 40 years [2], being the most common therapeutic class for its treatment in early ages [3].

Fluoxetine is one of the SSRI-class antidepressants, with oral formulations ranging from 20 mg to 90 mg. Its metabolism is mainly performed in the liver by the highly polymorphic CYP2D6. The contribution from other CYPs, such as CYP2C19, CYP2C9, and CYP3A4, is relevant in subjects with chronic exposure to fluoxetine since the drug inhibits its metabolism [4]. The pharmacogenomics of antidepressants is of interest given 48% therapeutic failures, 40% remission rate, and 25% frequency of adverse events [3,5]. Understanding the pharmacokinetic and pharmacodynamic genes’ contribution to SSRI function is currently an active research area, with above 900 studies reported [3]. Some of these studies have found solid clinical evidence, enough to generate regulatory guidance for pharmacogenomic testing for treatment selection and dose adjustments, such as those of the Food and Drugs Administration (FDA) and European Medicines Agency (EMA). 

International pharmacogenomics groups, such as the Clinical Pharmacogenetics Implementation Consortium (CPIC), have developed recommendations and calculator tools to predict the phenotype of actionable genes for CYP2D6 and CYP2C19, main cytochromes on the metabolism of antidepressants [6,7]. These tools, together with clinical evidence, allow clinicians to improve the therapeutics of antidepressants. Nevertheless, details and validation across different populations are advisable to strengthen the clinical evidence.

Bioequivalence trials can ensure quality to speed up the registration of generic medications by several regulatory agencies across the world, such as the FDA, EMA, and the Mexican regulatory agency, COFEPRIS [8]. In these studies, pharmacokinetic (PK) profiles are evaluated between test and reference formulations to compare their bioavailability, i.e., to test whether the geometric ratios of the PK parameters are—unless otherwise stated—inside an 80–125% confidence interval. Since two-sequence two-period two-formulation crossover trials are the most convenient designs for bioequivalence evaluation, intrasubject (within-subject) variability is required to estimate a sufficient sample size for assessing the confidence intervals with targeted error and statistical power.

In the world, there are 394 protocols reported in clinicaltrials.gov with the active principle fluoxetine, ranging from phase 1 to 4, 12 of them PK or bioequivalence studies [9]. In the Mexican National Registry of Clinical Trials (RNEC), there are 15 protocols registered with this drug [10]. Of those, 13 are bioequivalence/comparative bioavailability studies, the 20 mg capsule formulation being the most frequently studied. These studies, overall, had a crossover two-sequence two-period design, with a sample size ranging from 24 to 36 subjects. 

In this work, we report a clinical trial with fluoxetine 20 mg capsules performed in 24 healthy Mexican volunteers, where the pharmacokinetics was evaluated in a crossover randomized study, exploring relationships with the genotype of the participants. We aimed to test bioequivalence between two fluoxetine formulations, and to associate the genotype of participant subjects with the metabolizing phenotypes (pharmacokinetics) of fluoxetine, quantifying the potential impact of selected CYPs in the metabolism of fluoxetine. So, the bioequivalence evaluation was independent from the pharmacogenetic analyses performed for exploratory purposes.

## 2. Methods

### 2.1. Bioequivalence Clinical Trial

The clinical trial was a randomized controlled, open-label, crossover 2 × 2 (two-sequences, two-periods) design, with a single administration per period in fasting conditions in healthy subjects. Products administered were fluoxetine hydrochloride 20 mg capsules, Apotex, Inc. Toronto, ON, Canada (test product), and Prozac^®^ 20, Eli Lilly and Company (reference product). Blood sampling was performed up to 72 h after drug administration, with an 8-week washout between periods.

The sample size for the bioequivalence trial was based on an expected intrasubject variation of 18% for fluoxetine PK parameters [11], α of 0.05, and a power of 90% (β of 0.10), with 80–125% acceptance limits for the 90% confidence intervals, evaluated with Schuirmann’s two one-sided *t*-tests (TOST), in compliance with standard bioequivalence analysis methodology and national regulation [8,12]. This calculation resulted in a minimum sample size of 22 + 4 subjects due to potential dropouts.

Inclusion criteria were age from 18 to 55 years, body mass index (BMI) between 18 and 27 kg/m^2^, and clinical examination and laboratory data within normal ranges. Exclusion criteria were pregnancy or lactation; history of alcoholism, drug abuse, or heavy smoking; documented hypersensitivity to the study drug or to any drug in the same therapeutic group; receiving any medication within less than 14 days (or 7 half-lives) of the study starting date; subjects who had donated blood or had more than 450 mL of blood withdrawn 60 days before the study starting; history or evidence of medical conditions including, but not limited to, gastrointestinal, renal, hepatic, cardiovascular, and endocrine diseases; and consumption of grapefruit juice and/or xanthine-containing beverages, such as coffee, tea, chocolate, cola soft drinks, etc., 10 h before the study start-up.

The study was conducted in a bioequivalence laboratory facility in Mexico City, Axis Clinicals Latina, authorized as a clinical and bioanalytical unit for bioavailability and bioequivalence studies according to national requirements [8]. The clinical trial protocol was reviewed by an institutional review board (IRB) authorized by COFEPRIS and the National Commission of Bioethics (CONBIOETICA); then the protocol was submitted to the federal agency COFEPRIS for its approval.

All participant volunteers met the inclusion criteria and signed written informed consent. The study was conducted under NOM-177-SSA1-2013 [8], Good Clinical Practices (GCPs) [13], International Council for Harmonisation of Technical Requirements for Pharmaceuticals for Human Use (ICH) Guidelines [14], and followed the statutes of the Declaration of Helsinki and its amendments [15].

### 2.2. Plasma Fluoxetine Determination

The blood samples were collected in sodium heparin tubes at 0.0, 1.0, 2.0, 3.0, 4.0, 5.0, 6.0, 7.0, 8.0, 9.0, 10.0, 12.0, 24.0, 48.0, and 72.0 h post-dose. Once collected, samples were centrifuged at 3500 rpm for 10 min between 2 and 15 °C to separate plasma. Plasma was pipetted into labeled cryogenic tubes for fluoxetine determination. Liquid–liquid extraction and high-performance liquid chromatography coupled to mass/mass spectrometry (LC-MS/MS) with reverse phase column (Gemini, C18, 50 × 4.6 mm, 5 µm) was performed for fluoxetine quantification. Chromatographic conditions were 3 min run at 35 °C, with acetonitrile/ammonium formate 2 mM pH 3.5 (90:10 *v*/*v*) mobile phase, in turbo-electro spray ionization (ESI^+^) positive mode with multiple reaction monitoring (MRM), using paroxetine as internal standard.

### 2.3. Genetic and Pharmacogenomic Analyses

These analyses were funded and performed at the National Institute for Genomic Medicine (INMEGEN), with microarrays purchased from Código 46. The white blood cells’ buffy coat was pooled into a cryotube for DNA extraction for each subject, using Puregene blood kit (QIAGEN^®^), from 3 to 4 blood samples collected for PK to avoid low concentration of genetic material or poor quality. Quantity and quality were verified with spectrophotometry and agarose gel electrophoresis. Genotyping was performed using the Infinium^®^ Global Screening Array (GSA) 24 v1.0 microarray, including 600,000 variants with more than 22,000 customized clinically relevant genetic variants related to drug response [16].

### 2.4. Pharmacokinetics and Statistical Analysis

After fluoxetine determination, the formulation administered in each period for each subject was decoded according to the study randomization. Blood sampling actual times were calculated based on time deviations reported to the planned nominal sampling times. Observed plasma profiles vs. actual times were plotted and their PK parameters were calculated using non-compartmental analysis (NCA) in Phoenix^®^ WinNonlin^®^ version 8.1 [17]. PK parameters included maximum drug concentration (C_max_), time to C_max_ (t_max_), area under the curve from pre-dose to last sampling time (AUC_0–t_), AUC from pre-dose extrapolated to infinite (AUC_0–∞_), elimination parameters such as elimination constant (K_el_), half-life time (t_1/2_), and apparent distribution volume (Vd).

Statistical analysis for bioequivalence assessment was performed using Schuirmann’s two one-sided *t*-test (TOST) for test/reference geometric means ratios of C_max_ and AUC, applying the conventional acceptance criteria of 80–125% [12]. Its 90% confidence intervals were calculated following NOM-177-SSA1-2013 regulation. Outlier analysis was performed to identify subjects with extreme values, based on the studentized intrasubject residuals using bear package, BE/BA for R [18]. Analysis of variance (ANOVA) with a general linear model (GLM) for bioequivalence crossover studies was performed [12] to evaluate period, sequence, and formulation effects.

### 2.5. Genotyping and Genotype-Phenotype Association

The pharmacogenetic analyses were performed for scientific research purposes, since bioequivalence was assessed for all the subjects who completed the clinical study, as described in the previous section in accordance with the national regulation [8]. Analysis of the genomic data was carried out with Illumina^®^ GenomeStudio 2.0 software, performing quality control, normalization, and allelic discrimination of the variants of the 24 volunteers who concluded the bioequivalence study. Individual fluoxetine PK parameters were used as phenotypic variables. Multivariate association models were generated from data in plain text files with genetic information (.map and .ped), demographic variables (gender, age, weight, height, and BMI), and PK data (C_max_, AUC_0–t_, AUC_0–∞_, t_max_, t_1/2_), using gPLINK vs2.050. Supporting analyses were performed with IBM SPSS^®^ 25 [19] and Phoenix^®^ WinNonlin^®^ v.8.1, for the identification of the relevant genetic variants and their association with fluoxetine PK [20,21].

## 3. Results

### 3.1. Bioequivalence Clinical Trial

Twenty-six healthy Mexican subjects were recruited, assessed as healthy by physical examination and blood test, and started the study. All enrolled volunteers were tested for abuse drugs and alcohol consumption before being checked into each of the study periods. Twenty-four of them completed the study because two subjects (ID numbers 6 and 21) dropped out due to personal decisions and withdrew their informed consent (Figure 1). Demographic characteristics of the 24 subjects, 20 males and 4 females, were 33.0 ± 11.1 years old (mean ± SD), ranging from 19 to 50 years. Height was 1.65 ± 0.09 m, weight 64.8 ± 9.1 kg, and BMI 23.8 ± 2.1 kg/m^2^. Between genders, there was no difference in age (*p* = 0.855) and BMI (*p* = 0.157), although differences in weight and height were found (*p* < 0.000). Among these four demographic variables (age, BMI, height, and weight), we found significant correlations between age and BMI, age and weight, BMI and weight, and height and weight (*p* < 0.05).

A total of six adverse events in five subjects were reported, from which five events occurred under reference product dosing and one with the test product dosing. All adverse events were assessed as mild or moderate in severity. Four were classified as probably related to the drug due to occurrence around fluoxetine t_max_ (dizziness, headache, sore throat, and pharyngitis). The other two adverse events were not related to the study drug (lumbar pain at the discharge of period 1 and headache at period 2 check-in).

### 3.2. Plasma Fluoxetine Determination

The bioanalytical methodology for fluoxetine quantification was validated under the national regulatory agency criteria applicable to bioequivalence trials [8]. Validation parameters included accuracy, repeatability, reproducibility, robustness, selectivity, recovery, and stability, analyzed for both analyte (fluoxetine) and internal standard (paroxetine). The calibration curve of fluoxetine had a concentration range of 0.200–49.169 ng/mL (r = 0.99984), with low, medium, and high control samples at 0.598, 20.467, and 37.513 ng/mL. The coefficients of variation of the validation parameters were calculated within the acceptable regulatory values, all below 15%. The selectivity of the method was evaluated with common drugs, such as salicylic acid, metamizole, ketorolac, paracetamol, and ondansetron, demonstrating no interference with the analyte of interest.

### 3.3. Pharmacokinetic Parameters and Bioequivalence Testing

The average pharmacokinetic profile for each product is shown in Figure 2. In the PK analysis of fluoxetine (Table 1), comparing test and reference products, the mean t_max_ was 4.42 h and 4.25 h (*p* = 0.75), while mean C_max_ was 16.25 ng/mL and 15.92 ng/mL (*p* = 0.63). Half-life t_1/2_ was 30.71 h and 31.32 h (*p* = 0.42). The averages of AUC_0–t_ and AUC_0–∞_ were not statistically different (*p* = 0.64, *p* = 0.56), nor were Vd and Cl elimination parameters (*p* = 0.26, *p* = 0.34).

Bioequivalence statistical assessment was performed with the PK data of the 24 volunteers who completed the study (Table 2), confirming bioequivalence between test and reference products, since the 90% confidence intervals of the PK parameters (C_max_, AUC_0–t_, and AUC_0–∞_) were within the 80–125% limits. No statistically significant effects of sequence (*p* > 0.42), period (*p* > 0.12), and formulation (*p* > 0.08) were found in the ANOVA.

Four subjects (IDs 7, 11, 14, and 24) were identified to have extreme values in the PK parameters, based on ±2 studentized intrasubject (within-subject) residual criteria. Regarding intersubject (between-subject) variability, four volunteers had C_max_ values close to or above ±2 S.D. (IDs 4, 9, 11, and 23), as well as two subjects in the AUC parameters (subjects 11 and 25) (Figure 3). Phenotypically, 33% of the 24 subjects who concluded the bioequivalence study (IDs 4, 7, 9, 11, 14, 23, 24 and 25) were of interest given their high intra- and/or intersubject variability on fluoxetine PK parameters.

Mean PK profiles were different between male and female subjects, the PK parameters C_max_ (*p* < 0.000), AUC_0–t_ (*p* < 0.002), Vd (*p* < 0.001), and Cl (*p* < 0.006) being statistically different between genders. It is important to notice that the female population is underrepresented in this study due to recruitment, so gender differences should be taken as rough evidence. After analyzing the PK parameters (C_max_, AUC_0–t_, AUC_0–∞_, t_max_, and t_1/2_) with demographic variables, significant correlations (*p* < 0.05) were found between age and C_max_, weight and C_max_, weight and AUC_0–t_, height and C_max_, and height and AUC_0–t_. No statistically significant correlation (*p* > 0.146) was obtained for BMI vs. any of the PK parameters. Correlation coefficients for all explored variables are shown in Table 3. Variables with the strongest correlation, C_max_ vs. weight, are plotted in Figure 4, showing a negative trend for both genders and overall.

### 3.4. Pharmacogenomic Analyses

The 24 subjects that completed the study were genotyped. All studied single nucleotide variants (SNV) had a call rate > 0.99 and p10GC (gene call 10th percentile) > 0.30. Using a candidate gene approach [22], we selected SNVs of enzymes implicated in drug metabolism, especially those known to be involved in the metabolism of fluoxetine. A total of 423 potential variants of eight initial cytochromes of interest were investigated in PharmGKB (*CYP1A2*, *CYP2B6*, *CYP2C8*, *CYP2C9*, *CYP2C19*, *CYP2D6*, *CYP3A4*, and *CYP3A5*). From those, we found 169 coincidences in the customized microarray using a script developed in R and these were manually revised for duplicate removal. In total, 39 variants of 8 CYPs were studied: 5 for *CYP1A2*, 9 for *CYP2B6*, 1 for *CYP2C8*, 5 for *CYP2C9*, 6 for *CYP2C19*, 7 for *CYP2D6*, 5 for *CYP3A4*, and 1 for *CYP3A5*. Additionally, 15 pharmacodynamic and antidepressant response variants were explored in 4 genes: 3 for *ABCB1*, 3 for *SLC6A4*, 2 for *TPH1*, and 7 for *COMT*. Table 4 summarizes the genes and SNVs with clinical annotations and evidence levels available in the current literature [23,24]. 

Based on the sample of Mexican subjects participating in this trial, the minor allele frequencies (MAF) were similar to data reported for different populations in most of the SNVs studied. Excluding the SNV with MAF = 0% in our study, the following SNVs showed a difference greater than 10% with respect to the calculated populations’ mean (Table 5): *ABCB1* rs2032582: 43% higher; *CYP1A2* rs2069514: 20% higher, similar to Latino/admixed American (AMR) frequency; *CYP2B6* rs4802101: 16% lower; *CYP2D6* rs108098: 13% higher, similar to AMR frequency.

### 3.5. Association of PK Parameters with Pharmacogenomic Variants

Table 6 summarizes the individual variant significant additive allele effect (ADD) on the PK parameters as response variables. Covariates gender and age were significant (*p* < 0.05) in the models for C_max_ in the SNVs for *CYP2B6*, *CYP2D6,* and *ABCB1*; age was significant (*p* < 0.05) for AUC_0–t_ and AUC_0–∞_ in the *CYP2B6* and *CYP2D6* SNV models; and gender was significant (*p* < 0.05) in *CYP2B6* SNVs for Cl, and AUC_0–t_ in *CYP2B6* and *CYP2D6* SNVs. Only significant variants are shown in Table 6; some covariates might have significance, such as age and gender in C_max_ for *CYP2C19* and *SLC6A4* SNVs, and gender in AUC_0–t_ for *CYP2C19*, *CYP1A2*, and *CYP3A4* SNVs.

We performed multiple linear regression analysis with response variables *y* = pharmacokinetic parameter vs. *x*, being multiple SNVs, including demographic covariables. Appendix A displays the statistically significant models (*p* < 0.000), calculated using a stepwise method with the selected SNVs. *CYP2D6* SNVs are present in all regression models, and the correlation was reduced with the inclusion of gender variable in the models for the Cl parameter, and with the introduction of height in the AUC_0–t_ model. Significant *CYP2D6* SNVs were rs1065852, rs1135840, rs28371706, and rs28371703, the latter being in all regression models with high coefficients. Four *CYP2C19* SNVs were found significant: rs11188072, rs12769205, rs4244285, and rs4917623.

Other CYP SNVs such as *CYP1A2* rs2470890 are significant in the regression models, as well as *CYP2C9* SNVs rs1799853, rs28371686, and rs2256871. *CYP3A4* variants also seem to be relevant for PK parameters, with five different associated variants. From all the statistical models executed, out of the total 54 SNVs initially explored from 12 genes, 39 SNVs resulted in significance for 10 genes: 3 for *CYP1A2*, 6 for *CYP2B6*, 3 for *CYP2C9*, 4 for *CYP2C19*, 6 for *CYP2D6*, 5 for *CYP3A4*, 1 for *CYP3A5,* 3 for *ABCB1*, 2 for *SLC6A4*, and 1 for *TPH1.*

The *CYP2D6*10* haplotype, related to decreased CYP2D6 function (poor drug metabolism), was found in only one volunteer (subject 25), based on *CYP2D6* rs1065852 (haplotype A/A) and rs11358490 (haplotype C/C). As shown in Table 2, t_1/2_ was three times higher (106.9 h) for this subject than the overall mean (31.02 h). AUCs and t_1/2_ were statistically different between genotypes of *CYP2D6* rs1065852 (*p* < 0.001). Stratification of subjects based on *CYP2D6* genotypes confirmed the difference in the PK profiles, based on the three SNVs found in this study (rs1065852, rs1135840, and rs28371703) (Figure 5).

After removing from the bioequivalence analysis the PK data of the subject with decreased CYP2D6 function based on *CYP2D6 *10*, which was an extreme outlier in the statistical analysis and identified phenotypically as a poor metabolizer of fluoxetine, there was a reduction of 4.22% in the intersubject variability in AUC_0–t_ (44.97% vs. 40.75%) and of 13.66% in AUC_0–∞_ (57.12% vs. 43.46%) (Table 7).

## 4. Discussion

Despite having better therapeutic efficacy than first-generation and other third-generation antidepressants, the clinical response to fluoxetine is still highly variable, with 10.6% of patients discontinuing treatment for major depressive disorder due to a lack of therapeutic efficacy [28]. Because of that, pharmacological research on fluoxetine is relevant since the lack of therapeutic efficacy could derive from pharmacokinetic causes. Fluoxetine exhibits high intersubject PK variability, as confirmed in our study. Also, the exploration of different ethnicities is important for a better understanding of the pharmacogenomics of antidepressants, assessing both their pharmacokinetics and efficacy [29,30,31].

In this bioequivalence study in a Mexican sample, fluoxetine t_1/2_ was calculated as 30.7 h and 31.3 h for the test and reference formulations, respectively. These values are lower than those reported in the international literature for the drug, ranging between 4 and 6 days [32]. Nevertheless, our obtained values match with those reported for extensive metabolizers [33]. Maximum t_1/2_ values in our study sample were 88.9 and 106.9 h (3.7 and 4.5 days), corresponding to a subject with poor CYP2D6 metabolizer phenotype. Mean t_max_ was found around 1 h before the reported value, confirming a high proportion of extensive metabolizers in the sample studied, which has also been found by other researchers in the Mexican population [34].

Bioequivalence was concluded between test and reference formulations of fluoxetine 20 mg capsules with a 2 × 2 crossover design, in which each subject serves as its own control. Maximum intrasubject variability was 14.4% for C_max_, so the sample size of 24 subjects was enough to achieve the minimum acceptable statistical power of 80%. Nevertheless, intersubject variability was high, up to 57.1% for AUC_0–∞_, which is also well known [35]. It has been previously reported that drugs mainly metabolized by CYP3A4 tend to have higher intrasubject PK variability, while drugs primarily metabolized by CYP2D6 show high intersubject variability in general [36], which is confirmed for fluoxetine in this study.

The quantitative impact of CYP variants is reported in this study. *CYP2D6* rs28371703 had the greatest impact on the PK parameters of fluoxetine in the regression models. One subject with homozygote allele A/A for SNV rs1065852, associated with decreased CYP2D6 function, was found in our study sample, and we confirmed that the plasmatic t_1/2_ increased about three times compared with A/G and G/G alleles. This SNV frequency in our study (^1^/_24_ = 4.2%) is consistent with reports in the literature, ranging from 2.8% and up to 4.3% in Caucasian, Hispanic, and Afro-American populations [37]. Together with *CYP2D6* rs11358490, this SNV is part of the *CYP2D6*10* haplotype [7], known to have decreased cytochrome function along with other gene variants [38]. Moreover, a phenotype–genotype relationship was identified for three *CYP2D6* variants, confirming the relevance of this cytochrome in fluoxetine metabolism. CYP2C19 activity was found to be associated with differential exposure to the drug based on the genotype, as well as CYPs *1A2*, *2C9*, and *3A4*, confirming previous findings [39]. Six *CYP2B6* SNVs were found relevant in fluoxetine metabolism. CYP2B6 is also known to participate in antidepressant metabolism [40], so attention should be also pointed to this cytochrome when administering SSRI, as recently discussed in CPIC guidelines [41].

In this study, we identified 34 significant gene variants on 10 genes with quantifiable relevance in the pharmacokinetics of fluoxetine, with a candidate gene approach. Out of those, 29 SNVs belong to 7 CYPs (*CYP1A2*, *CYP2B6*, *CYP2C9*, *CYP2C19*, *CYP2D6*, *CYP3A4*, and *CYP3A5*), and 5 SNVs to 3 pharmacodynamics (PD)- and efficacy-related genes (*SLC6A4*, *TPH1*, *ABCB1*). These three genes encoding for the serotonin transporter 1, *p*-glycoprotein 1, and L-tryptophan hydroxylase 1, respectively, are directly involved in the PD of fluoxetine. These have been researched thoroughly in the past [42], aiming for a better understanding of the efficacy of antidepressants from the PD perspective. Even though they seem important in fluoxetine PK, the role of PK-related proteins in drug PD is multifactorial, and medication response still cannot be associated purely with single gene variations, as has been explored and evidenced in similar studies for other therapeutic classes [43].

Regarding the limitations of this study, it is important to note that the sample size required for bioequivalence studies is considerably smaller than that required for pharmacogenetic studies, so we recommend more research into pharmacogenetics in pharmacokinetic studies, considering pharmacogenetics as an exploratory objective. Also, further evidence should be gathered to confirm the differences in fluoxetine PK between genders, as the number of females in this study was considerably lower than males. Finally, this study was conducted in healthy volunteers to explore fluoxetine PK, without the intention of evaluating its efficacy.

With this clinical study with fluoxetine hydrochloride, we confirm that through genotyping participating subjects in bioequivalence studies, the drug exposure can be assessed, and significant reduction of the intersubject variability can be achieved in clinical trials. The latter was evidenced despite the limitation of the sample size for the pharmacogenetic analyses within the context of a bioequivalence study for determining interchangeability between two fluoxetine formulations.

## Figures and Tables

**Figure 1 jpm-13-01352-f001:**
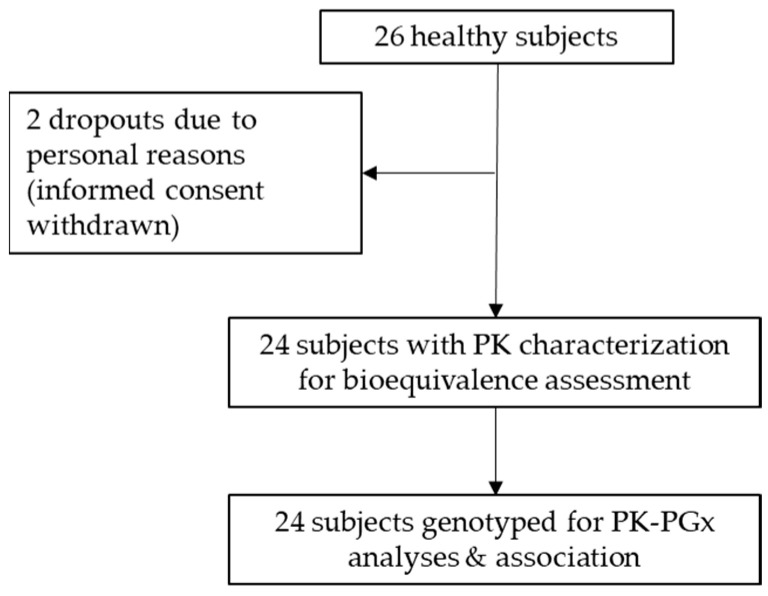
Study flow diagram.

**Figure 2 jpm-13-01352-f002:**
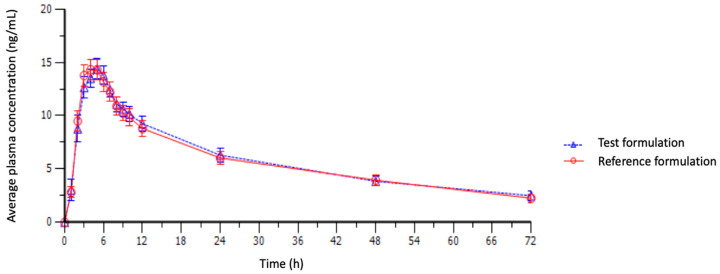
Mean plasma concentration–time curve of fluoxetine after an oral dose of 20 mg (*n* = 24).

**Figure 3 jpm-13-01352-f003:**
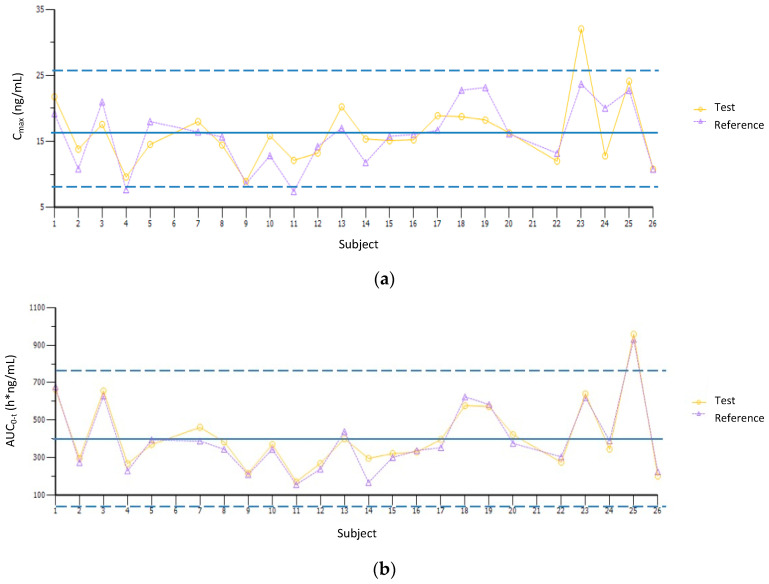
Individual fluoxetine PK parameters: (**a**) C_max_ and (**b**) AUC_0–t_. The solid blue line indicates the overall mean for each parameter, while the dashed lines represent ±2 S.D.

**Figure 4 jpm-13-01352-f004:**
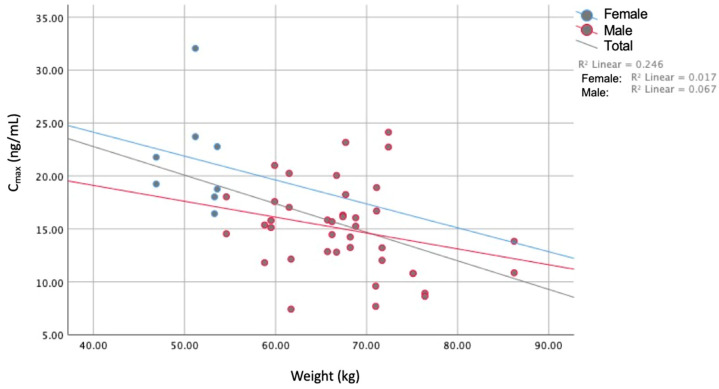
Weight vs. C_max_, stratified by gender and overall.

**Figure 5 jpm-13-01352-f005:**
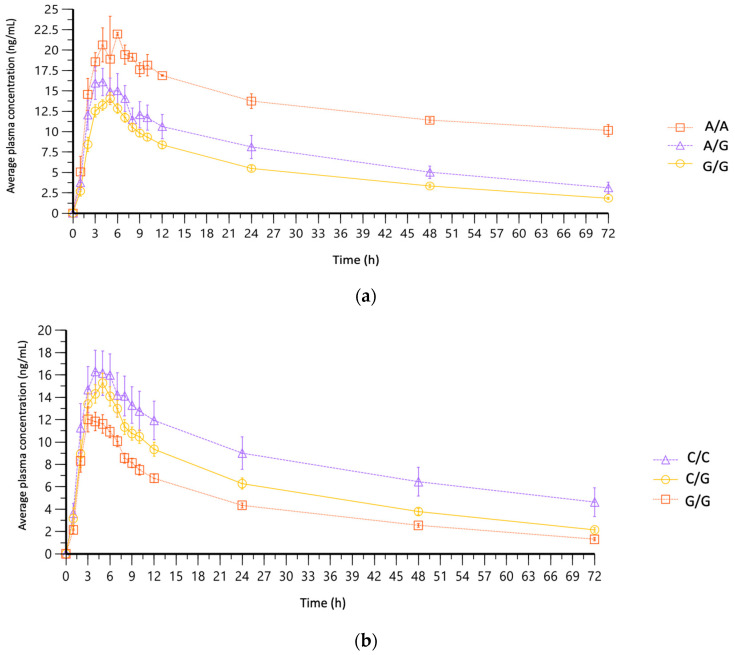
Mean plasma concentration–time curve of fluoxetine, stratified by *CYP2D6* genotypes: (**a**) rs1065852 (100C>T), associated with decreased function *CYP2D6*10* allele (A/A *n* = 1; A/G *n* = 3; G/G *n* = 20); (**b**) rs1135840 (4180G>C), associated with decreased function *CYP2D6*10* allele (C/C *n* = 4; C/G *n* = 13; G/G *n* = 7); (**c**) rs28371703 (974C>A), associated with no function *CYP2D6*4* allele (A/C *n* = 4; C/C *n* = 20).

**Table 1 jpm-13-01352-t001:** Fluoxetine pharmacokinetics (PK) parameters.

	Test Product					Reference Product				
	C_max_	AUC_0–t_	AUC_0–∞_	t_max_	t_1/2_	C_max_	AUC_0–t_	AUC_0–∞_	t_max_	t_1/2_
Mean	16.25	411.64	556.06	4.42	30.71	15.92	398.19	542.43	4.25	31.32
Geometric mean	15.59	377.86	473.26	4.25	28.90	15.12	360.06	457.47	4.07	29.02
Standard deviation (S.D.)	5.02	184.05	429.46	1.18	13.78	4.89	189.55	436.65	1.19	17.21
Standard error of the mean (SEM)	1.02	37.57	87.66	0.24	2.81	1.00	38.69	89.13	0.24	3.51
Minimum	8.92	171.86	200.73	2.00	15.42	7.42	157.33	171.14	2.00	19.24
Median	15.31	371.42	452.21	5.00	27.04	16.10	349.68	442.86	4.50	27.60
Maximum	32.05	957.96	2351.79	6.00	88.86	23.71	930.62	2383.39	6.00	106.88
Coefficient of variation (%)	30.90	44.70	77.20	26.60	44.90	30.70	47.60	80.50	28.00	55.00

Units: C_max_ [ng/mL], AUC_0–t_ and AUC_0–∞_ [h*ng/mL], t_max_ and t_1/2_ [h].

**Table 2 jpm-13-01352-t002:** Fluoxetine bioequivalence assessment (*n* = 24).

Parameter	Geometric Mean		T/R Ratio (%)	90% Confidence Interval	Intrasubject Variability (%)	Intersubject Variability (%)	Power (%)
	Test (T) Product	Reference (R) Product					
C_max_	15.59	15.12	103.10	96.05–110.67	14.4%	29.27%	100.0%
AUC_0–t_	377.85	360.05	104.94	99.87–110.28	10.0%	44.97%	100.0%
AUC_0–∞_	473.25	457.47	103.45	100.23–106.77	6.4%	57.12%	100.0%

**Table 3 jpm-13-01352-t003:** Demographic and PK variables correlation matrix.

		BMI	Age	Weight	Height	C_max_	AUC_0–t_	AUC_0–∞_	t_max_	t_1/2_
**BMI**	Pearson Correlation	1	0.481 *	0.665 **	0.023	−0.213	−0.179	−0.020	0.103	0.133
	*p*		0.017	0.000	0.913	0.146	0.223	0.891	0.484	0.368
**Age**	Pearson Correlation		1	0.323 *	−0.014	−0.344 *	−0.250	−0.209	0.252	0.015
	*p*			0.123	0.950	0.017	0.087	0.153	0.084	0.921
**Weight**	Pearson Correlation			1	0.758 **	−0.496 **	−0.359 *	−0.113	0.003	0.129
	*p*				0.000	0.000	0.012	0.446	0.981	0.380
**Height**	Pearson Correlation				1	−0.483 **	−0.340 *	−0.141	−0.078	0.056
	*p*					0.001	0.018	0.339	0.598	0.705
**C_max_**	Pearson Correlation					1	0.825 **	0.609 **	−0.158	0.301 *
	*p*						0.000	0.000	0.285	0.037
**AUC_0–t_**	Pearson Correlation						1	0.895 **	0.069	0.676 **
	*p*							0.000	0.640	0.000
**AUC** ** _0–∞_ **	Pearson Correlation							1	0.086	0.902 **
	*p*								0.559	0.000
**t_max_**	Pearson Correlation								1	0.158
	*p*									0.282
**t_1/2_**	Pearson Correlation									1
	*p*									

* Correlation significance < 0.05 (two-tailed). ** Correlation significance < 0.01 (two-tailed).

**Table 4 jpm-13-01352-t004:** Genes and single nucleotide variants (SNVs) in the study with available annotations [23,24].

Gene	SNV	Associated Haplotype	Genetic Variation	Annotated Variants	Clinical Annotations	Evidence Level (CPIC)	Classification	Related Drug	Function
*CYP1A2*	rs2069514	** 1C*	3860G>A	17	2	3	N/A	Antipsychotic	Efficacy
*CYP1A2*	rs2069526	** 1K*	739T>G	7	1	3	Intronic	Escitalopram	Toxicity
*CYP2B6*	rs4802101		750T>C	3	1	3	5′ end	Cyclophosphamide	Toxicity
*CYP2C19*	rs11188072			8	2	3	N/A	Escitalopram	Dose
*CYP2C8*	rs11572080	** 3*		58	14	3	Missense	Rosiglitazone	
*CYP2C9*	rs28371686	** 5*	42619C>G	23	5	1A	Missense	Phenytoin	Metabolism
*CYP2D6*	rs1080985	** 2A*	1496C>G	3	2	3	3′ end	Thioridazine	Efficacy
*CYP2D6*	rs1135840	** 4, * 10*	4180G>C	828	82	3	Missense	Fluoxetine	
*CYP2D6*	rs16947	** 21*	2851C>T	643	77	3	Missense	Fluoxetine	
*CYP2D6*	rs72549358	** 28*		2	1	4	Intronic	Tamoxifen	
*CYP3A4*	rs2687116		C>A	2	0	N/A	Intronic	--	
*ABCB1*	rs1045642		G>A	580	100	3	Missense	SSRI	Efficacy
*ABCB1*	rs1128503		1236T>C	216	38	3	Syno-nym	Methadone	Metabolism/PK
*ABCB1*	rs2032582		2677T>G/A	315	49	3	Missense	Fluoxetine	Efficacy
*SLC6A4*	rs1042173		A>C	2	2	3	3′ UTR	Ondansetron	Efficacy
*SLC6A4*	rs2066713		G>A	1	1	3	Intronic	Ethanol	Toxicity
*SLC6A4*	rs25532			12	3		N/A		
*TPH1*	rs1799913		G>T	1	0	N/A	Intronic	Lithium	Efficacy

**Table 5 jpm-13-01352-t005:** Minor allele frequencies (MAF) of analyzed genes from all available (ALL), African American (AFR), East Asian (EAS), South Asian (SAS), European (EUR), Latino/admixed American (AMR) populations, from ALFA dbSNP, and the sample in this study (MXN) [25,26,27].

Gene	rs	Allele	ALL	AFR	EAS	SAS	EUR	AMR	MXN
*CYP1A2*	rs2069526	G	0.071	0.123	0.084	0.069	0.024	0.025	0.063
*CYP1A2*	rs2069514	A	0.209	0.313	0.281	0.080	0.020	0.362	0.413
*CYP2B6*	rs4802101	T	0.246	0.057	0.340	0.252	0.414	0.221	0.083
*CYP2C19*	rs11188072	T	0.156	0.246	0.015	0.136	0.224	0.120	0.083
*CYP2C8*	rs11572080	T	0.046	0.008	0.001	0.030	0.118	0.091	0.065
*CYP2C9*	rs28371686	G	0.005	0.017	0.000	0.000	0.000	0.001	0.021
*CYP2D6*	rs72549358	T	0.001	0.000	0.000	0.000	0.005	0.001	0.021
*CYP2D6*	rs1080985°	C	0.162	0.051	0.103	0.240	0.241	0.239	0.292
*CYP2D6*	rs16947°	A	0.368	0.568	0.135	0.357	0.338	0.332	0.326
*CYP2D6*	rs1135840°	C	0.404	0.322	0.292	0.474	0.460	0.530	0.438
*CYP3A4*	rs2687116	C	0.220	0.721	0.005	0.045	0.030	0.102	0.083
*ABCB1*	rs1045642	A	0.395	0.150	0.398	0.575	0.518	0.428	0.438
*ABCB1*	rs1128503	A	0.416	0.136	0.627	0.587	0.416	0.403	0.458
*ABCB1*	rs2032582	T	0.049	0.001	0.134	0.050	0.018	0.059	0.479
*SLC6A4*	rs2066713	A	0.257	0.254	0.066	0.289	0.381	0.314	0.292
*SLC6A4*	rs1042173	C	0.485	0.185	0.822	0.552	0.437	0.542	0.375
*TPH1*	rs1799913	T	0.321	0.163	0.475	0.269	0.391	0.372	0.354
Sample size			5008	1322	1008	978	1006	694	24

Data from 1000Genomes_30x, with a global *n* = 6404 genomes (1786 AFR; 1170 EAS; 1202 SAS; 1266 EUR; 980 AMR).

**Table 6 jpm-13-01352-t006:** Additive allele effect (ADD) of SNVs associated with PK parameters.

PK Parameter	Gene	Chromosome	SNV	Minor Allele	Coefficient (ADD)	*p*-Value
C_max_	*ABCB1*	7	rs2032582	C	−2.27	0.033
	*ABCB1*	7	rs1128503	G	−2.11	0.042
	** *CYP2B6* **	**19**	**rs4802101**	**A**	**5.38**	**0.009**
	*CYP2B6*	19	rs4803418	G	−2.90	0.010
	*CYP2B6*	19	rs4803419	A	−2.90	0.010
	*CYP2B6*	19	rs2279344	G	4.94	0.005
	*CYP2D6*	22	rs28371703	A	4.87	0.016
AUC_0–t_	*CYP2B6*	19	rs4802101	A	219.30	0.022
	*CYP2B6*	19	rs4803418	G	−156.30	0.002
	*CYP2B6*	19	rs4803419	A	−156.30	0.002
	*CYP2B6*	19	rs2279344	G	180.50	0.033
	*CYP2D6*	22	rs1135840	C	142.00	0.005
	** *CYP2D6* **	**22**	**rs28371703**	**A**	**291.80**	**0.001**
	*CYP2D6*	22	rs1065852	A	249.80	0.001
AUC_0–∞_	*CYP2B6*	19	rs4802101	A	568.30	0.020
	*CYP2B6*	19	rs4803418	G	−336.90	0.010
	*CYP2B6*	19	rs4803419	A	−336.90	0.010
	*CYP2B6*	19	rs2279344	G	463.50	0.032
	*CYP2D6*	22	rs1135840	C	301.80	0.023
	*CYP2D6*	22	rs28371703	A	733.80	0.001
	** *CYP2D6* **	**22**	**rs1065852**	**A**	**783.60**	**0.000**
t_max_	** *CYP2C9* **	**10**	**rs2256871**	**G**	**−1.99**	**0.047**
t_1/2_	*SLC6A4*	17	rs1042173	A	9.89	0.049
	*CYP2B6*	19	rs4802101	A	18.43	0.045
	*CYP2B6*	19	rs2279344	G	16.82	0.035
	*CYP2D6*	22	rs28371703	A	23.54	0.006
	** *CYP2D6* **	**22**	**rs1065852**	**A**	**28.64**	**0.000**
Cl	*ABCB1*	7	rs2032582	C	14.20	0.018
	*ABCB1*	7	rs1128503	G	11.77	0.048
	*CYP2B6*	19	rs4802101	A	−24.28	0.047
	*CYP2B6*	19	rs4803418	G	14.00	0.035
	*CYP2B6*	19	rs4803419	A	14.00	0.035
	*CYP2B6*	19	rs2279344	G	−22.78	0.031
	** *CYP2D6* **	**22**	**rs28371703**	**A**	**−26.79**	**0.022**
	*CYP2D6*	22	rs1065852	A	−21.36	0.043

Only SNVs with *p* < 0.05 are summarized. SNVs with the lower *p*-value are written in bold text.

**Table 7 jpm-13-01352-t007:** Fluoxetine bioequivalence assessment after excluding the PK data of the slow-metabolizing subject (*n* = 23).

Parameter	Geometric Mean		T/R Ratio (%)	90% Confidence Interval	Intrasubject Variability (%)	Intersubject Variability (%)	Power (%)
	Test (T) Product	Reference (R) Product					
C_max_	15.32	14.83	103.28	95.89–111.23	14.7%	28.37%	100.0%
AUC_0–t_	363.16	346.11	104.93	99.61–110.53	10.3%	40.75%	100.0%
AUC_0–∞_	441.64	426.27	103.61	100.24–107.09	6.5%	43.46%	100.0%

## Data Availability

The authors declare that the data supporting the findings of this study are mostly available within the paper. Complementary data are available upon request via email to the authors.

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
