# Peer review of "Pharmacogenetic Variants Associated with Fluoxetine Pharmacokinetics from a Bioequivalence Study in Healthy Subjects"

_jpm, 2023, doi:10.3390/jpm13091352_

Round 1

Reviewer 1 Report

The article studies a current topic for the “Journal of Personalized Medicine”, precisely the pharmacogenetic features determining the pharmacokinetic behavior of fluoxetine, even though the basic is the bioequivalence study. In general, the article sounds technically very good, but there are some minor issues, that may not have become clear and the authors might be able to explain.

1.     It is mentioned that 24 subjects were included in the study out of 26, but it is not mentioned which exactly dropped out (as a numbers).

2.     The graphs (in Figure 2) show values of 26 subjects and note the numbers of subjects (subjects 4, 7, 9, 11, 14, 23, 24 and 25) of interest for high intra- and/or inter-subject variability on fluoxetine PK parameters. It can be assumed that all of them are from those who have completed the study. Is that correct? It should be clarified.

3.     It is written that mean PK profiles were different between male and female subjects (line 199). Isn't the number of females in the study too small to draw this conclusion?

4.     Among the subjects of interest mentioned above (subjects 4, 7, 9, 11, 14, 23, 24 and 25), only subject 25 is discussed about the reduced activity of CYP2D6. For the rest it is not clear why there is no information and explanation

Author Response

Dear reviewer,

We are greatly thankful for your time, thorough review, and comments on our article. Please see the attachment with a detailed response to each one of your comments and suggestions.

Kind regards,

The authors.

Reviewer 2 Report

The manuscript is dealing with investigation of the pharmacogenetic variants associated with fluoxetine pharmacokinetics from a bioequivalence study in healthy subjects. The manuscript require revision concerning the below comments and suggestions to be performed.

1-      Study flow chart is missed, to be submitted.

2-      Sample size calculation was not performed. This raised a concern about the accuracy of the results which were based on the number of the volunteers.

3-      Inclusion and exclusion criteria to be added in the methodology.

4-      Bioanalytical method validation was not mentioned in the manuscript. This raised a concern about the accuracy and precision as well as the sensitivity of the analysis method.

5-      Page 3 line 119 PK and statistical analysis” to be corrected to “Pharmacokinetics and statistical analysis”

6-      Page 4 lines 170 to 172 ; the following mentioned paragraph “Possible concomitant medications during the study, such as salicylic acid, metamizole, ketorolac, paracetamol, and ondansetron, were evaluated to confirm the selectivity of the method, demonstrating no interference with the analyte of interest.” require a scientific clarification and explanation which include concurrent medications which might be administered by the volunteers although it is well known that criteria to be fulfilled in bioequivalence studies according to FDA guidelines that volunteers should not administer any prescribed or OTC medication before the study by about 30 days and during the study to the end of the study.

7-      The aim of the study was to test bioequivalence between two fluoxetine formulations, and to associate the genotype of participant subjects with the metabolizing phenotypes (pharmacokinetics) of fluoxetine, quantifying the impact of selected CYPs in the metabolism of fluoxetine. This raised a concern about the bioequivalence study part and its significance on the genotyping part which are not correlated to each other as pharmacokinetics study was enough and there was no need for bioequivalence study.

8-      In the results the authors detect the Association of PK parameters with pharmacogenomic variants title. Concerning the pharmacokinetics parameters are correlated to reference or test product or to both? Then what was the significance of undergoing bioequivalence study here?

9-      The study was performed on healthy volunteers and not on patients, this to be written as study limitation.

10-  Recommendation of the study is required. 

11-  The following paragraph recommended to be added in the introduction or discussion showing similar study performed;

         It was reported a clinical study for investigation of the association between food/UGT2B7 polymorphisms and pharmacokinetics/pharmacodynamics properties of indapamide in healthy humans that food independently decreased the value of indapamide’s Tmax  and increased the value of Cmax and that all genetic variants of both UGT2B7 SNPs showed a non-significant impact on the values of Tmax, Cmax, and AUC0–t; however, it was found that rs11740316 variant AG was correlated with a 2.8-h lower MRTinf. Moreover, the participant’s body mass index was positively correlated with longer MRTinf.

https://doi.org/10.3390/biomedicines11051501

Author Response

(The authors gave the same response as above.)

Round 2

Reviewer 2 Report

The required modifications, suggestions were performed as well as the raised comments were done, 

The quality of the manuscript was significantly improved, good Luck!